# A deep learning framework for the localization of landmarks on the lateral semi circular canals

Zhixuan Wei[ID][1]*, Sudanthi Wijewickrema[ID][1,2], Bridget Copson[1,3], Jean-Marc Gerard[1,4], Stephen O'Leary[1,4]

1 Department of Surgery (Otolarynglogy), University of Melbourne, Melbourne, Australia, 2 Australian Centre for Artificial Intelligence in Medical Innovation, La Trobe University, Melbourne, Australia, 3 Department of Radiology, St Vincent's Hospital, Melbourne, Australia, 4 Department of Otology, Royal Victorian Eye & Ear Hospital, Melbourne, Australia

* wezw@student.unimelb.edu.au

## Abstract

This paper introduces a Deep Learning (DL) framework to localize landmark coordinates within the semicircular canals in Computed Tomography (CT) scans of the temporal bone. These landmarks can be consistently defined across patients and imaging modalities and as such can serve as a means of forming a common coordinate system. We propose a DL based framework for automating the landmark selection process. We establish the accuracy of the methods using Bone Beam CT scans of the temporal bone of 20 patients and landmarks selected by 3 human experts as the ground truth. We show that the error rates are similar to the levels of variation in landmark selection achieved by human experts. We further validated the method on CT scans from 14 additional patients, demonstrating that the accuracy remains within clinically acceptable parameters.

## Introduction

Landmark localization, the identification of points or regions of interest in an image, is an important task in many computer vision applications. In medical image analysis, it typically involves the identification of anatomical landmarks. These landmarks are then used for tasks such as clinical measurement [1], definition of clinical coordinate systems [2], and registration [3]. Traditionally, identification of anatomical landmarks has fallen to experts such as radiologists and surgeons. However, this is a cumbersome and time consuming process that often introduces a level of subjectivity, due to factors such as task complexity and level of experience of the expert. Automated landmark localization, typically based on machine learning techniques, provides a faster and more robust alternative.

Two main machine learning approaches have been used in automated landmark localization: traditional machine learning techniques and deep learning. The former uses features extracted from the image [4] to train a recognition model such a neural

which permits unrestricted use, distribution, and reproduction in any medium, provided the original author and source are credited.

**Data availability statement:** Data cannot be shared publicly because of [ethical restrictions]. Data are available from the Royal Victorian Eye and Ear Hospital Institutional Data Access / Ethics Committee (contact via the author Dr Zhixuan Wei - wezw@student.unimelb.edu.au, Pro Stephen O'Leary - sjoleary@unimelb.edu.au, the head of the Department of Otolaryngology and also the owner of the ethics (#08/796H/18) of the data, researchers can also contact Kerryn Baker (kerryn.baker@eyeandear.org.au), Education Precinct Manager of the Royal Victorian Eye and Ear Hospital, about the ethics) for researchers who meet the criteria for access to confidential data.

**Funding:** The author(s) received no specific funding for this work.

**Competing interests:** The authors have declared that no competing interests exist.

network or support vector machine [5]. In contrast, deep learning (DL) techniques are able to learn the ideal features for the task directly from the input data as part of the training process [6,7].

In this project, we aim to automatically identify four landmarks within the lateral semicircular canals (LSCC). Previous work has established that these landmarks can be viewed clearly in both CT and MR imaging of the temporal bone and can be identified with a high level of accuracy and reproducibility [8]. They were chosen for the specific purpose of defining a patient-specific coordinate system that can be used for planning skull base surgery. The precision required for this type of surgery, is around 0.3 mm-0.5 mm [9].

We use Deep Learning methods to localize these landmarks in 3D CT scans. We introduce a multi-stage framework that takes advantage of the level of detail available in high resolution images, yet does not require the same high levels of memory and processing power. To this end, we first train a simple DL model on images that have been rescaled to a lower resolution to identify the rough landmark locations. We then determine the optimal crop size based on these locations using an exhaustive approach. Finally, we train and compare several DL methods on cropped images of the original resolution. We select the best performing methods and tune/optimize them to achieve better, more robust results. We show that a high level of accuracy can be achieved in the detection of the LSCC landmarks of Copson et al. [8]. We further show that the error rates are well within the levels of precision required for skull base surgery and are similar to the variation seen between the landmarks selected by human experts.

## Previous work

Numerous DL methods have been introduced to address different 3D medical image landmark localization tasks. For example, Li et al. [10] built a framework using the attention-guided mechanism to train models for anatomical landmark localization on CT scans. They defined 5 landmarks around the condyle such as external auditory meatus apex and lowest point of the articular nodule. The framework contained two stages: the first stage trained a model using low resolution scans to identify the approximate landmark coordinates, and the second stage used the rough locations to crop out patches from high resolution scans and used these patches to train models. The authors stated that the patches can focus the model on the landmarks and reduce the effects of unrelated area. The models used in the framework shared the same architecture of Ronneberger et al., which consisted of a 3D U-Net [11] followed by a Differentiable Spatial Numerical Transform (DSNT) [12] layer. They reported the mean and standard deviation of error distances of these 5 landmarks as: 2.56(2.68), 1.98(1.96), 2.13(1.84), 4.67(4.45) and 2.07(2.02) in mm.

Payer et al. [13] investigated the use of regressing heatmaps for multiple landmark detection and experimented with several models which included Down-Sampling Convolutional Neural Network (CNN), Convolution Only Network (ConvOnly), U-Net and SpatialConfiguration-Net (SCN). The SCN was a novel model introduced in this paper. They evaluated the localization performance on two different

datasets consisting of images of the hand: 2D X-ray images and 3D MRI scans. For the 3D task, 28 annotated landmarks such as finger tips and bone joints we used. The results showed that U-Net and SCN achieved the best median landmark localization precision (1.10 mm). They further achieved mean and standard deviation of 1.18(1.31) mm and 1.19(1.48) mm respectively. For the 2D task where 37 landmarks were identified, the best results they achieved was: median of 0.64 mm, mean of 0.85 mm and standard deviation of 1.01 mm for the model in Lindner et al. [14].

Yeh et al. [6] constructed a Fully Convolutional Network model to detect 45 anatomical landmarks on a dataset of 2D whole-spine lateral radiographies. Their model was based on Cascaded Pyramid Network (CPN) [15], which comprised two outputs in one model that predicted the results from coarse to fine. The first part of the model was a U-shaped structure similar to a U-Net, but with Residual Blocks [16] added during the Down-Sampling and Up-Sampling processes. It produced a rough prediction as the first output. The second part refined the blocks from the Up-Sampling process of the first part using multiple residual blocks and concatenated them to produce a more precise prediction. Both of the predictions were used in the Wing loss function [17] to optimize the model. The major difference between their model and the CPN was that they used a DSNT layer on the heatmap of each stage to obtain the final landmark coordinate output. The median localization error distance of all the landmarks ranged from 1.75 to 3.39 mm.

Gajowczyk et al. [18] proposed a one-step coronary ostia landmark localization model for Computed Tomography Angiography (CTA) volumes. The model consisted of a residual U-Net with heatmap matching and DSNT. They applied the Jensen-Shannon divergence regularization with the popular Euclidean loss function during the training process. The purpose was to force the network to produce gaussian-like heatmaps, in order to improve the overall prediction, as stated in Nibali et al. [12]. The authors used 2 datasets in their study: an in-house dataset of coronary CTA (CCTA) and a public dataset of 3D CT Images (ImageTBAD) [19]. they detected 2 landmarks: the left and right coronary ostia. They reported the median Euclidean distance error on CCTA as 1.14 mm and 0.99 mm for the 2 landmarks respectively. On the ImageT-BAD dataset, the results were 3.48 mm and 2.97 mm respectively.

## Environment details

### Data

We used Cone Beam CT scans of the temporal bone from 20 patients with no cochlear or temporal bone pathology. We used CT scans of 14 patients not used in training or testing as an additional validation dataset. The mean volume size was 1020x1020x338 and the voxel size (resolution) was 0.15 mm in all 3 dimensions. Ethics approval for this was obtained from the Royal Victorian Eye and Ear Hospital Human Ethics Committee (#08/796H/18).

### Data collection

This was a retrospective study. The data were collected between January 2024 and November 2025. The researcher responsible for data collection had access to individual-level information. However, all data were de-identified prior to storage, and no identifying information was available thereafter.

### Landmarks of the LSCC

The four landmarks to be identified here were two identical points on the LSCC on either si de of the head: Left and Right Anterior and Posterior Landmarks. Fig 1 illustrates these. The Anterior Landmark was defined as the centre-point of the junction between the anterior aspect of the LSCC and the utricle – that is, the point at which the slender bony canal opens into the vestibule anteriorly – confirmed in all three imaging planes. The Posterior Landmark was defined equivalently at the posterior junction of the LSCC and the utricle, where the canal re-enters the vestibule posteriorly, again confirmed in all three planes. Three experts (2 consulting ear, nose, throat (ENT) surgeons and 1 radiologist) identified these 4 landmarks twice each on the CT scans of the 20 patients. The median coordinates of these 6 selections per landmark were

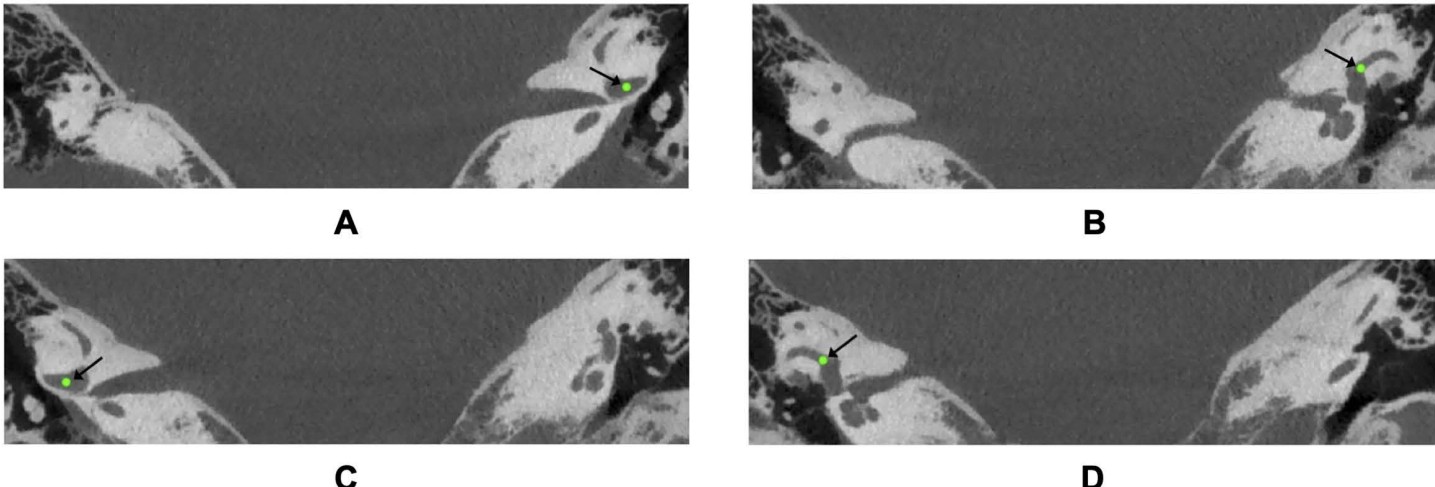

**Fig 1. Landmarks of the lateral semicircular canals. (A)** Left anterior LSCC landmark. **(B)** Left posterior LSCC landmark. **(C)** Right anterior LSCC landmark. **(D)** Right posterior LSCC landmark.

used as the ground truth to train and test models. The median was selected in order to reduce the subjectivity of selection and the effect of outliers.

## Data pre-processing

Since 4 landmarks consisted of 2 landmarks on either side of the head, we first simplified the problem from detecting 4 landmarks to 2. To this end, we divided the CT volumes into 2, along the middle slice in the sagittal direction, assuming that the scans were roughly in the same orientation. Then, we flipped the right half along the same plane so that all images were of the same (left) side of head. This effectively doubled the dataset (40 instances).

As this number of instances was still not sufficient to train Deep Learning models, and collecting more data was not practical in our case, we used data augmentation to supplement the dataset. In a review of augmentation techniques for medical data such as CT scans, Chlap, et al. [20] identified 4 broad classes of such methods: basic, deformable, deep learning, and other augmentation techniques. The method utlised here was random rotation and translation which falls under the category of basic augmentation. According to Chlap, et al. [20] 62% of the papers reviewed used basic augmentation techniques. The authors concluded that these techniques are effective in improving DL model performance. Following previous work ([21]), we rotated the volumes around a randomly chosen axis by a random angle ranging from −15 degrees to 15 degrees. Then we translated the volume by a random number of voxels (from −5 voxels to 5 voxels) along each of the axes X, Y and Z. We performed 50 such augmentations per instance to obtain a dataset of 2000 instances.

## Experimental setup

The programming language we used was Python, along with the Machine Learning packages Tensorflow and Keras. We used the Jupyter Notebook to visualize the results and Matlab to for data augmentation. All the models were trained on Spartan, the University of Melbourne High Performance Computing Cluster (single GPU: NVIDIA A100 80GB PCIe).

## Models

For the determination of landmark locations, regression-based DL models (consisting of feature extraction and inference parts) are typically used. We implemented several regression models based on previous work: Downsampling Network

(Down-Net), Shallow Downsampling Network (SDown-Net), Convolutional Only Network (ConvOnly-Net), Spatial Configuration Network (SCN), and Cascaded Pyramid Network (CPN). We combined the feature extraction parts of these models with 2 different inference methods: a dense (fully connected (FC)) layer and a Differentiable Spatial Numerical Transform (DSNT) [12] layer. The FC inference part consisted of a dropout layer, a flatten layer and a dense layer. The dropout rate was set to 0.2 and the number of units of the dense layer was set to 6 which is the number of target prediction coordinates (the number of landmarks (2) x number of coordinates (3)). The DSNT inference part consisted of a convolutional layer with a filter size equal to the number of prediction points (2) and a kernel size of 1, followed by a DSNT layer. Fig 2 shows the 2 types of inference used. The code is available at https://github.com/acherking/CT-MRI_LandmarkDetection/tree/dev.

## Downsampling network (Down-Net)

Downsampling architecture (Fig 3) involves down sampling functions like max-pooling (MAXPOOL) to reduce the size of the original input. The Down-Net we used was inspired by Islam, et al. [21] which was used in a registration task. The original network has 18 blocks, each of them comprising a 3D convolutional (CONV) layer, a ReLU activation function and a batch normalization (BN) layer. All the CONV layers use the same padding and have filters numbering from 32 to 1024. The majority have kernel sizes of 3 with some of size 1 in between. Among the 18 blocks, numbers 1, 2, 5, 8 and 13 have a MAXPOOL layer followed by a BN layer. Our implementation has 12 blocks, with all of the CONV layers having a kernel size of 3. They have a 32, 64, 128, 64, 128, 256, 128, 256, 512, 256, 512 and 256 filters respectively. Blocks 1, 2, 5, and 8 have a MAXPOOL layer. Fig 3 shows the architecture of this network.

## Convolutional only network (ConvOnly-Net)

In contrast to the Downsampling network, this architecture only uses CONV layers with a stride length of one and no pooling, to retain the resolution of the image. We constructed this structure based on Payer et al. [13]. Their network has

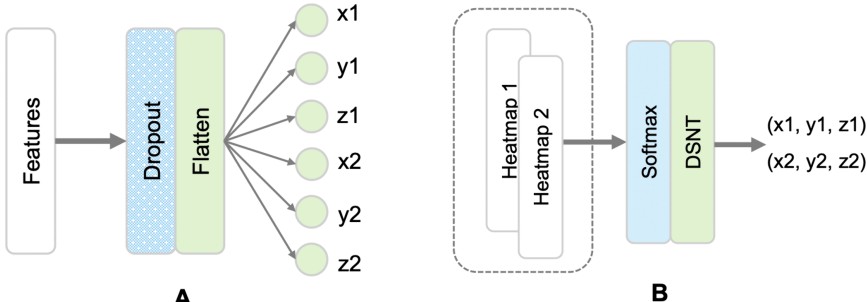

**Fig 2. Inference part of the landmark localization models. (A)** Fully connected inference. **(B)** DSNT inference.

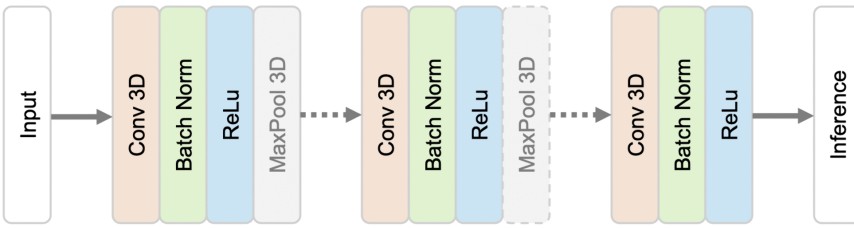

**Fig 3. Downsampling network (Down-Net).**

6 CONV layers using ReLU activation function. The kernel size is 5 and the number of filters is 32. In our implementation, we added a BN layer between the CONV layer and the ReLU function. We also increased the number of filters of the respective CONV layers to 32, 64, 128, 64, 128 and 64 respectively. We added a MAXPOOL layer with a kernel size of 4 for the FC inference to reduce calculation during the training process. See Fig 4 for an illustration of the ConvOnly-Net.

## U network (U-Net)

The U-Net was originally introduced by Ronneberger, et al. [11] for medical image segmentation. We implemented the original version which has a contracting path consisting of 4 down-sampling blocks and a symmetric expanding path consisting of 4 up-sampling blocks. There are 4 skip connections between the symmetric blocks. The down-sampling block has double CONV layers followed by a MAXPOOL layer and a dropout layer. There are two CONV layers with 1024 filters connecting the end of the contracting path and the start of the expanding path. The up-sampling block has a deconvolutional (DCONV) layer with a stride length of 2 followed by a dropout layer and double CONV layers. The number of filters are 64, 128, 256 and 512 in each block respectively. All the CONV/DCONV layers use a kernel size of 3 and same padding. The activation function is ReLU. We added an extra CONV layer at the end for each of the inference methods. For FC inference, the extra CONV has 3 filters, kernel size of 3, stride length of 4, and ReLU activation. For the DSNT inference, the extra CONV has 2 filters, kernel size of 1, same padding, and no activation function. Fig 5 shows the U-Net implementation.

## Spatial configuration network (SCN)

The SCN has three blocks: local appearance, spatial configuration and combination. The local appearance block contains four CONV layers with a kernel size of 5, same padding, and ReLU activation. The first 3 layers have 128 filters and the number of filters in the last layer is equal to the number of landmarks. First, the spatial configuration block uses average-pooling with a factor of 4 to down sample the input size. It then uses a CONV layer with 1 filter and a kernel size of (9, 9, 5) to produce a spatial configuration heatmap for each landmark using as inputs, the outputs of the local appearance block associated with the other landmarks. The combination block performs an element-wise (Hadamard) multiplication on the outputs of the local appearance and up-sampled spatial configuration blocks to produce the output features. We used this as the input for DSNT inference. In contrast, for FC inference, we down-sampled the output of the appearance block and used that and the output of the spatial configuration block to calculate the Hadamard product to reduce the number of calculations. Fig 6 illustrates this process.

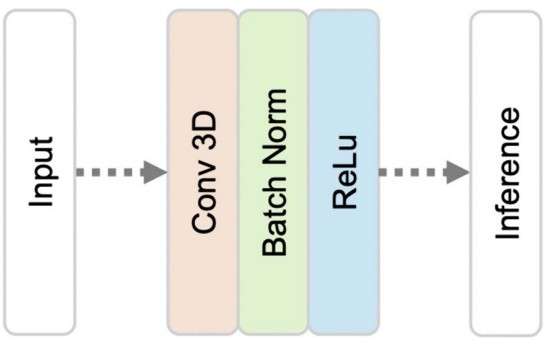

**Fig 4. Convolutional only network (ConvOnly-Net).**

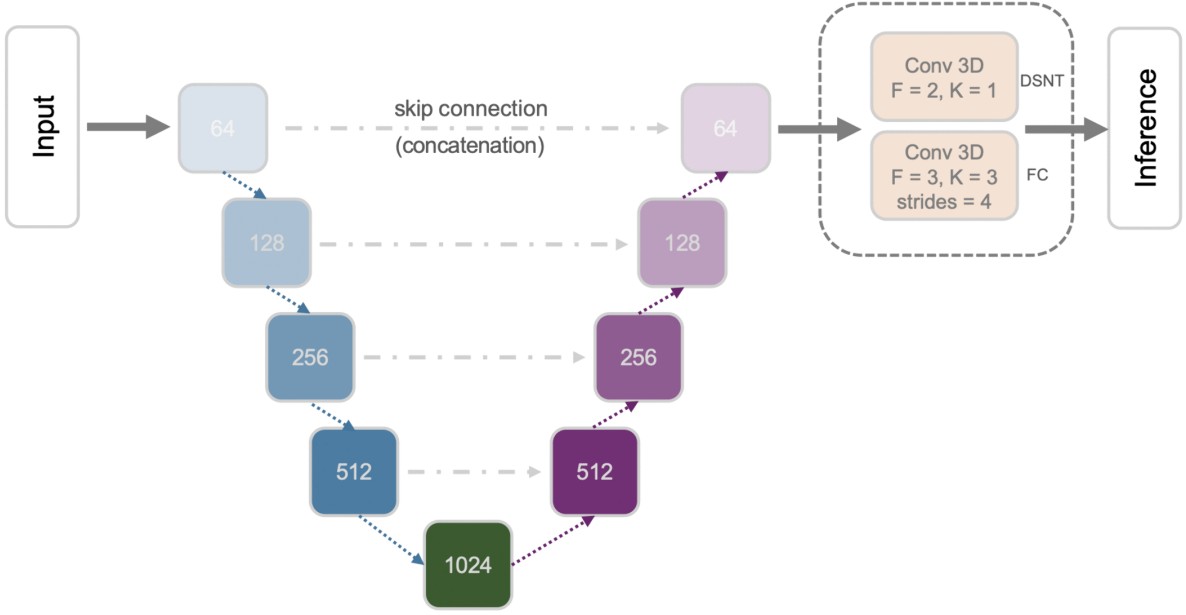

**Fig 5. U network (U-Net).**

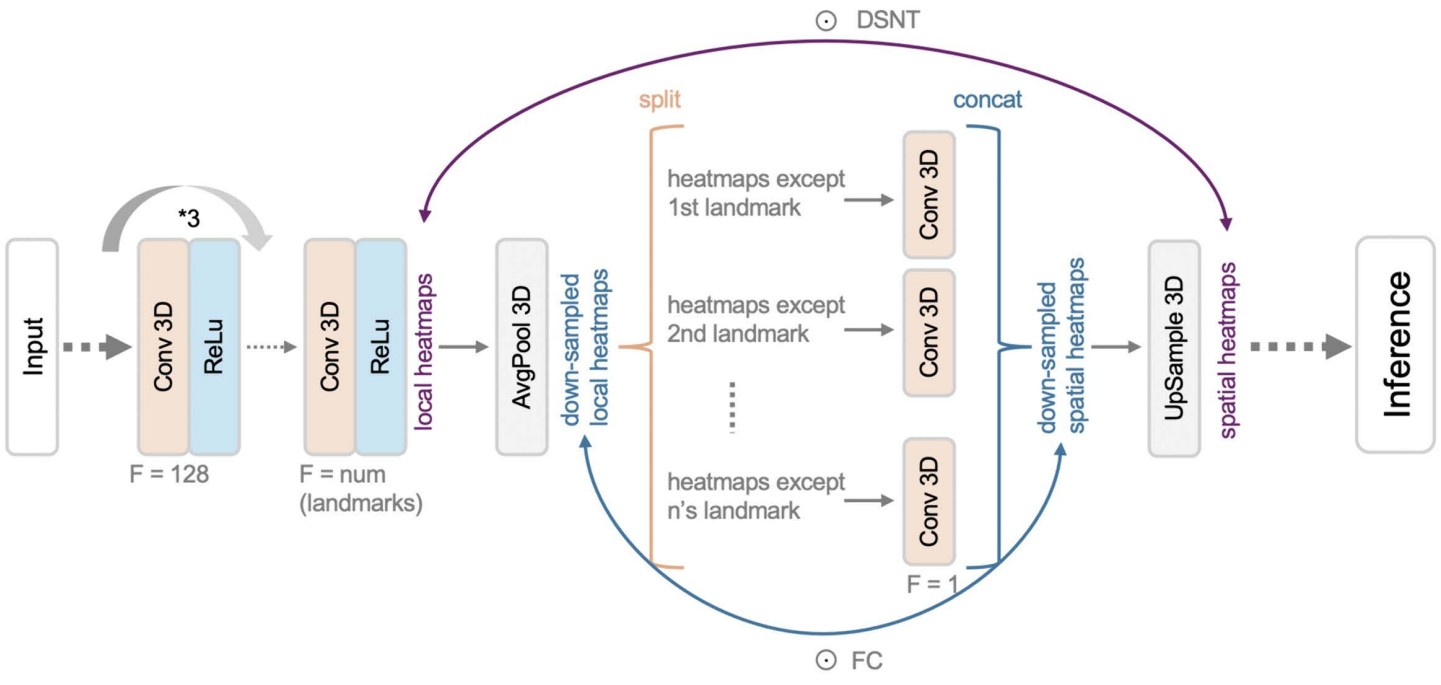

**Fig 6. Spatial configuration network (SCN).**

## Cascaded pyramid network (CPN)

We implemented the CPN based on Yeh et al. [6]. It has two stages and each produces a feature map. The general architecture of the first stage is similar to the U-Net which has a contracting path and an expanding path. Each path has three pairs of symmetric blocks. The output of each block of the contracting path is added to the output of the corresponding block in the expanding path. To down sample the input, instead of using double CONV and MAXPOOL, as in the original U-Net, it uses a residual block with 2 CONV layers, which have stride lengths of 2 and 1 respectively. The up-sampling function we used is 3D repetition rather than the linear function used in Yeh et al. [6]. In stage 2, the network applies several residual blocks on the 3 blocks in the expanding path of stage 1. It then concatenates the outputs together to produce another feature map. Fig 7 shows the architecture of the CPN.

## Landmark detection framework

The proposed landmark detection framework comprises 4 stages: 1) training of a DL model on down-scaled images to identify the approximate locations of the landmarks, 2) determining the region of interest (optimal crop-size), 3) using cropped images in the original resolution to train DL models to identify the landmarks at a high level of precision, and 4) robustness validation for selected models. Fig 8 illustrates this process.

## Determination of approximate landmark locations

After data augmentation and halving, the resulting volumes are around 1020x510x339 voxels in size. Training DL models for this size is impractical as it requires higher levels of memory and computational power. To avoid this, we rescaled all the instances to a fixed size of 176x88x48. Then, we trained a Down-Net using these downsized images to determine approximate locations of the landmarks. As this stage is to determine the approximate regions where the landmarks are located, we trained the model (Down-Net) to only detect the center of the anterior and posterior landmarks.

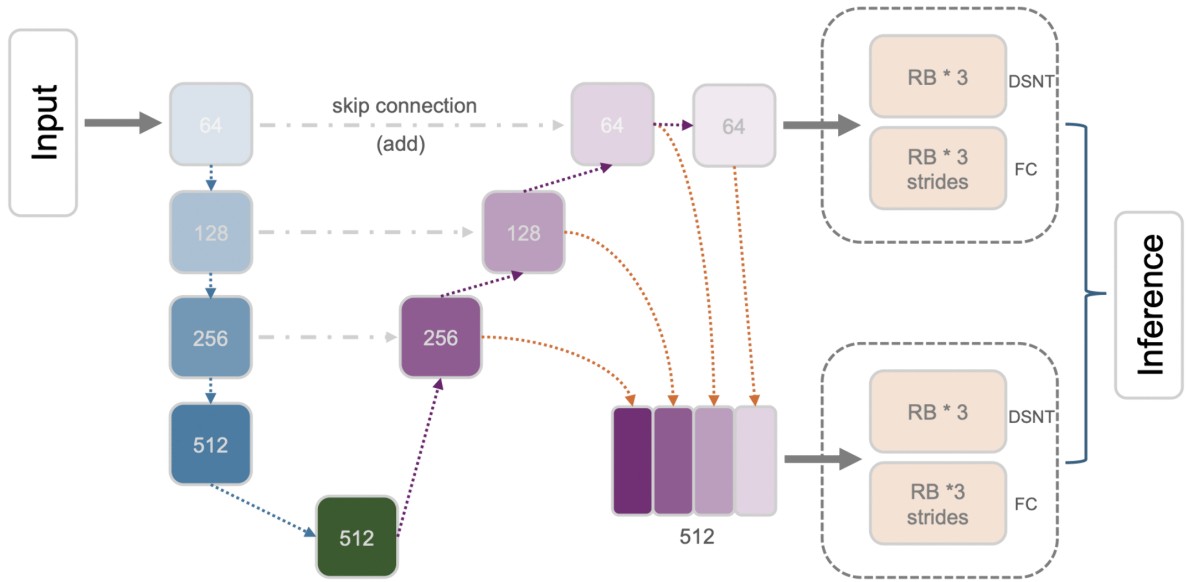

**Fig 7. Cascaded pyramid network (CPN).** RB: residual block.

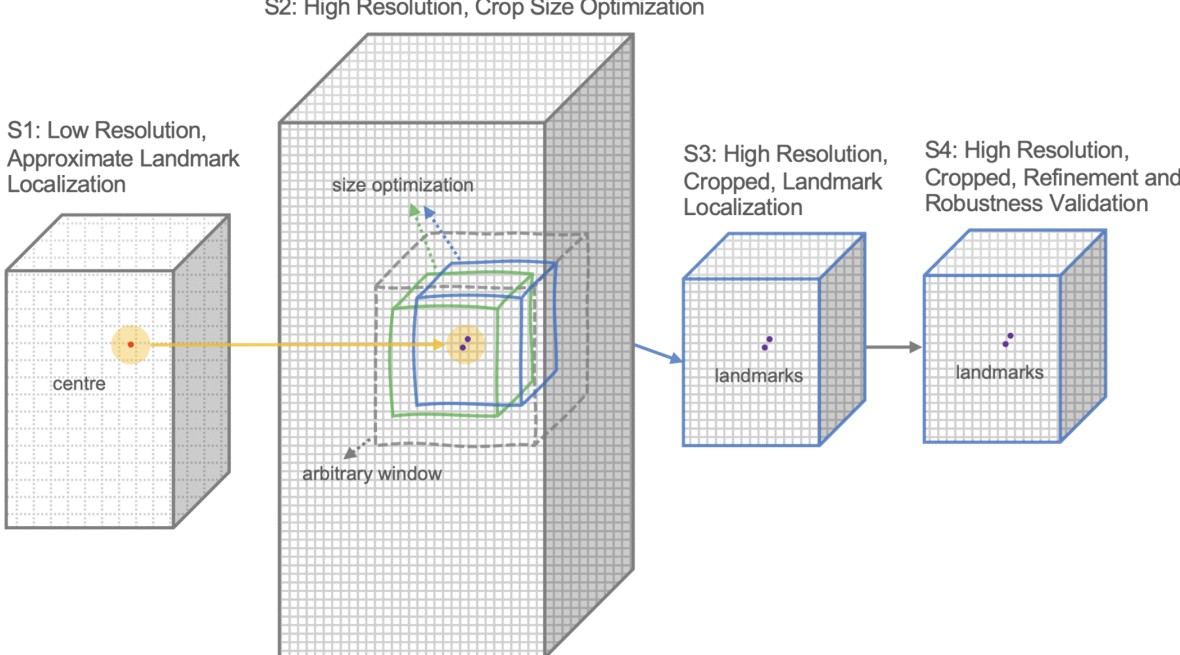

**Fig 8. Four-stage landmark detection framework.** At the first stage, the full images are resized and the approximate center of landmark locations is determined. In the second stage, this center is used as the center of a window of arbitrary size for crop size optimization. The third stage uses the optimal crop window (obtained from stage 2) to train deep learning models. In the fourth and final stage, the best models selected from the previous stage are fine-tuned for robustness validation.

However, as the augmentation process (the rigid body transformations of rotation and translation) preserves the resolution of the images but not their size, rescaling them into a fixed size results in images of different resolutions. To overcome this, we changed the error function to represent real-world distances (mm) by multiplying the voxel distances by the image resolution.

For this stage of the process, we divided the dataset into three subsets: 70% for training, 10% for validating and 20% for testing. To avoid bias, the data separation was performed so that each original image and all its augmentations were in the same subset. We used early stopping using the validation dataset to avoid overfitting. Fig 9 shows the training and validation results. The final test results for this stage (mean error) was 1.286 mm.

### Determination of region of interest

First, We used the predictions from the previous stage (the centre of the anterior and posterior landmarks), and converted it to the coordinates of the original images. Then we used it as the center to crop a region of size 100x100x100 from the original (higher resolution) images. We then trained a Down-Net with the cropped images. We used the size of 100x100x100 as it was large enough to include the landmarks and surrounding structures and provide sufficient information for DL models.

The next step was to determine the optimal crop size for this application. For this, we adopted an exhaustive approach. First, we assumed that if an outside layer of the image contributed little to the landmark detection process, replacing its intensity values with zeros will have little impact on the detection results. As such, we replaced a layer at a time of the image with zeroes moving from the border to the center, in one of 6 directions (row+, row-, column+, column-, slice+,

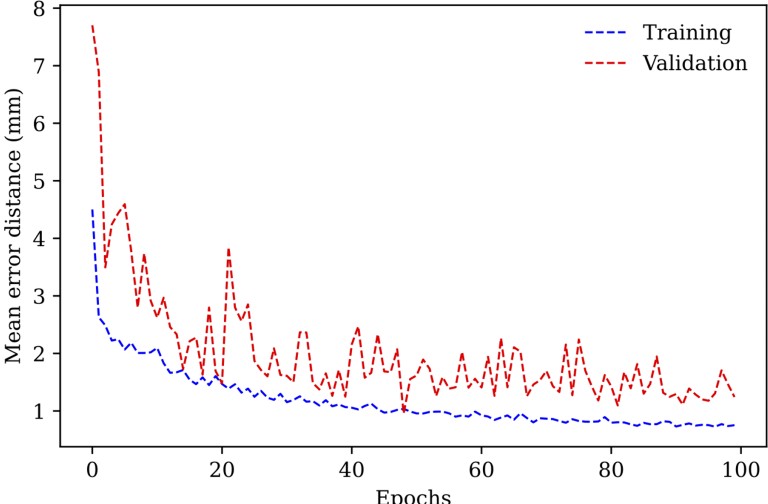

**Fig 9. Training and validation results for the approximate landmark detection stage.**

and slice-). For example, when considering the increasing row direction (row+), at iteration 6, a layer in the image from row = 1 to row = 6 was blanked out. Fig 10 illustrates this process.

We then observed the landmark prediction errors at each iteration in each of the 6 directions of the Down-Net trained on 100x100x100 sized images above. Fig 11 shows the changes in the prediction results with iterative replacement of the image layers. Next, we determined the layer depths for each direction to reach a given increase in the error (when compared to the original error of the model). The vertical bars in Fig 11 show the layer depths in each direction for error increases of 10% and 50%.

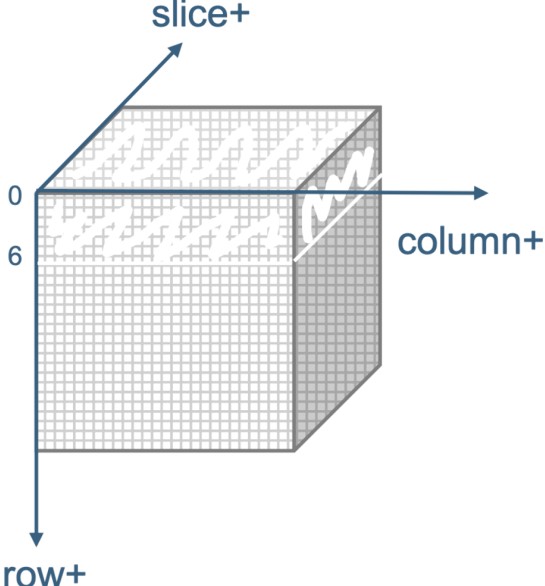

**Fig 10. Crop size detection process.** An example showing six outer layers removed from the image in the row+ direction.

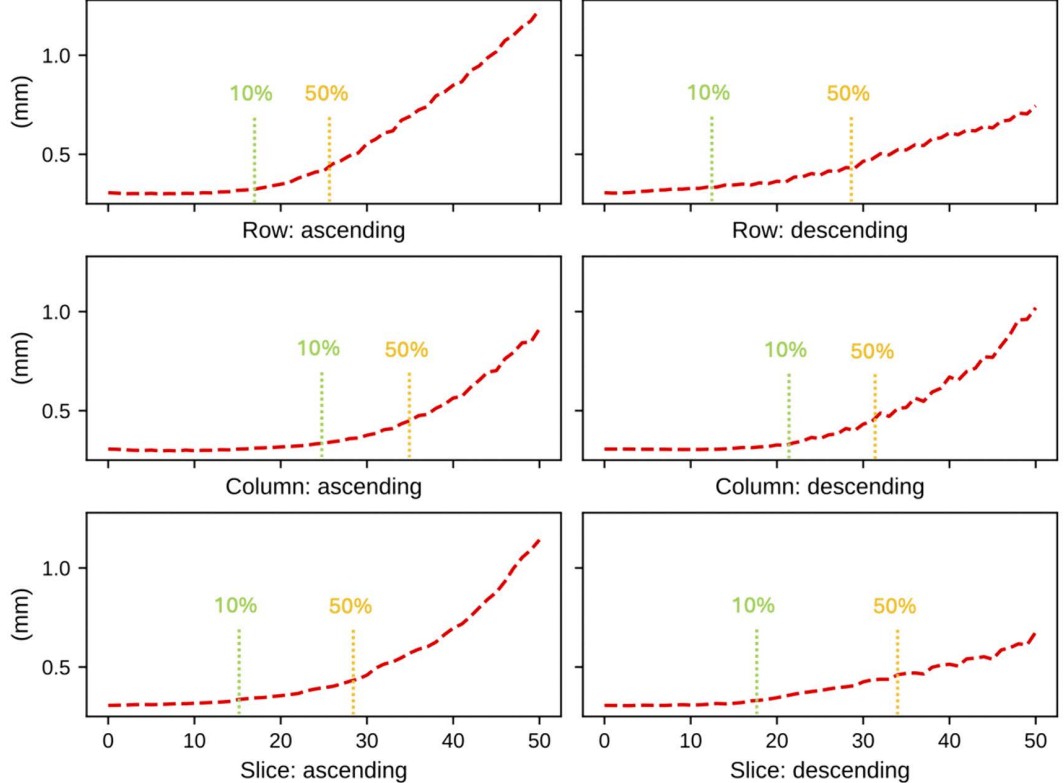

**Fig 11. Changes in prediction error with iterative layer removal.** Image layers were iteratively replaced with zero values in each of six directions: row+, row-, column+, column-, slice+, and slice-. The vertical bars represent the number of layers removed when the prediction error increased by 10% and 50% relative to the original prediction error.

The layer depths at a given error increase were then used to crop the 100x100x100 image to obtain a smaller image. We iteratively changed the increase in error from 10% to 50% in steps of 5% and trained a Down-Net with the corresponding crop sizes at each iteration. Fig 12 shows the results. We selected a crop size of 63x50x61 as this provided the best results.

## Determination of precise landmark locations

In this stage, we used the crop size determined in the previous stage to train the models discussed in Section Models. All the models were trained and optimized on the 70% training dataset and 10% validation dataset as above. The evaluations were assessed on the 20% blocked out test dataset. Tables 1 and 2 show the prediction errors of the models with fully connected and DSNT inference respectively.

## Hyper-parameter optimisation and cross validation

In the final stage, we used the results of the previous section to select 2 models for hyper-parameter optimization and robustness validation. We selected the Down-Net with FC inference and U-Net with DSNT inference as they showed the most accuracy and least variation in the test results. We further optimized these models using the random grid search in keras-tuner package. The hyper-parameters we optimized are the batch size, the optimizer, the learning rate and the decay steps. We used 8-fold cross validation to evaluate the robustness of the methods. To this end, we set aside 20% of

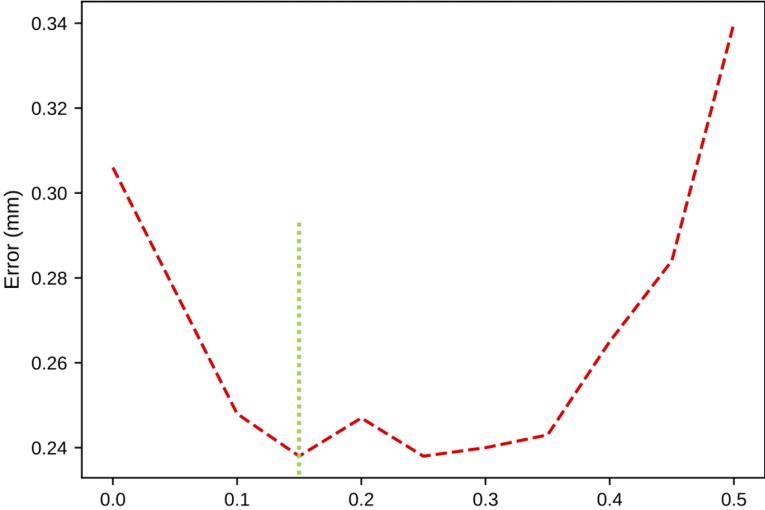

**Fig 12. Crop size optimization.**

**Table 1. Landmark localization results of the networks with fully connected inference. Best results are shown in bold.**

| Model | Weights | Training Time | Error (Mean (Std Dev)) mm | | |
|---|---|---|---|---|---|
| | | | Training | Validation | Test |
| Down-Net | 17.6M | 0.1h | 0.071 (0.040) | 0.172 (0.087) | **0.246 (0.128)** |
| SDown-Net | 4.8M | 0.08h | 0.143 (0.076) | 0.286 (0.158) | 0.381 (0.249) |
| ConvOnly-Net | 4.4M | 1.74h | 0.128 (0.080) | 0.293 (0.175) | 0.458 (0.290) |
| U-Net | 103.6M | 0.89h | 0.097 (0.043) | 0.267 (0.155) | 0.419 (0.253) |
| SCN | 4.2M | 2.07h | 0.206 (0.076) | 0.562 (0.256) | 0.760 (0.425) |
| CPN | 72.8M | 1.90h | 0.942 (1.082) | 1.013 (0.912) | 1.113 (0.964) |

**Table 2. Landmark localization results of the networks with DSNT inference. Best results are shown in bold.**

| Model | Weights | Training Time | Error (Mean (Std Dev)) mm | | |
|---|---|---|---|---|---|
| | | | Training | Validation | Test |
| Down-Net | 14.0M | 0.14h | 0.293 (0.457) | 0.346 (0.385) | 0.389 (0.417) |
| SDown-Net | 1.2M | 0.07h | 0.319 (0.446) | 0.414 (0.383) | 0.481 (0.408) |
| ConvOnly-Net | 4.4M | 2.18h | 0.133 (0.096) | 0.322 (0.190) | 0.430 (0.246) |
| U-Net | 103.5M | 0.89h | 0.059 (0.031) | 0.183 (0.100) | **0.290 (0.206)** |
| SCN | 4.1M | 2.04h | 0.084 (0.045) | 0.434 (0.213) | 0.470 (0.233) |
| CPN | 55.2M | 0.33h | 1.621 (0.782) | 1.524 (0.720) | 1.564 (0.807) |

the dataset, same as in earlier stages. We then divided the remaining 80% of the data into 8 subsets and at each iteration, used 7 of these for training and the remaining 1 for validation.

We used 60 combinations of randomly selected hyper parameter values and trained and validated 8 models at each iteration. We selected the hyper parameter values of the model with the best mean accuracy across the 8 folds. Then we retrained the model using the full training and validation dataset. Because at this stage, no validation was performed, we

used the mean number of epochs of the 8-fold training process to identify the early stop point during training. The final results were obtained as the performance of the tuned models for the test dataset which was set aside at the beginning. Table 3 presents the final results, the total error and error per landmark. It also shows the variation in the landmark coordinates detected by 3 human experts [8]. This was calculated as the mean (and standard deviation) of the distance between the median landmark coordinates and the coordinates selected by the experts.

### Additional validation

To further evaluate the reliability of the framework and demonstrate its potential clinical applicability, we tested it on an additional dataset which has not been seen by this framework. This dataset of CT scans from 14 patients were annotated by one expert clinician. We evaluated the framework's performance using the Wilcoxon Signed Rank test. Accuracy requirements in image guided surgery and image registration using landmarks are not standardised, and reasonable thresholds depend on the precision required by the surgical technique. For lateral skull base image guidance, a value of 0.5 mm has been suggested [22], and we adopted this as the predefined upper boundary for evaluating the clinical acceptability of our landmark detection results.

No augmentation was conducted for this validation. The models that performed the best on the test data: the optimized Down-Net with FC inference and the U-Net with DSNT inference, were used here. Figs 13 and 14 show the Bland-Altman plots comparing the models' predictions with the expert determined ground truth. The 95% limits of agreement (LoA), defined as the mean difference ±1.96 times the standard deviation of the differences, were used to quantify the range within which 95% of the individual differences between methods are expected to lie. Table 4 shows the euclidean distance between the models' prediction and the ground truth. Note the use of non-parametric metrics (such as the median and interquartile range) due to the small number of scans in the additional validation dataset, which prevents the assumption of normality required for parametric metrics. Prediction errors were compared against the suggested clinical upper boundary of 0.5 mm [22] using the Wilcoxon Signed Rank test to assess the statistical validity of the prediction results. Fig 15 visualises some prediction results.

### Discussion

Automating the identification of landmarks of the lateral semicircular canals enables patient scans to be transformed into a standardized coordinate system, improving the clinical workflow by (1) facilitating automation of downstream tasks such as multi-modal registration, surgical planning, and pathology identification, and (2) minimizing the need for human intervention.

We proposed a step-wise process for the detection of these landmarks and achieve a balance between accuracy and resource requirements. As seen from the final results of Table 3, the proposed methods achieve high levels of accuracy. These are comparable to the performance of DL methods addressing similar landmark detection tasks as discussed in Previous Work. The results are also well within the requirements for the application: lateral skull base surgery (0.3 mm-0.5 mm) [9]. Furthermore, the error rates are in the range of the variations present in the landmark selection of human experts [8]. However, the prediction of the posterior landmark was observed to be slightly less accurate than that of the anterior

**Table 3. Landmark location results of the optimised and cross validated models.**

| Model | Error (Mean (Std Dev)) mm | | |
| --- | --- | --- | --- |
| | Total | Anterior Landmark | Posterior Landmark |
| Down-Net (FC) | 0.239 (0.125) | 0.177 (0.096) | 0.301 (0.121) |
| U-Net (DSNT) | 0.288 (0.172) | 0.205 (0.122) | 0.371 (0.174) |
| Human Experts | 0.216 (0.244) | 0.219 (0.136) | 0.212 (0.318) |

**Fig 13. Bland–Altman plots for Down-Net with FC inference on the additional dataset.** Differences (in millimeters) between the model's predictions and the human expert's annotations are shown across all three axes. The bias (mean difference) is shown as a blue dashed line, and the limits of agreement (mean ± 1.96 × SD) are shown as red and green lines.

landmark. This could be because of higher variation in the ground truth for this landmark as seen in the larger standard deviation of the coordinates selected by human experts [8]. The reason behind the differences in human landmarks detections for anterior and posterior points may be the higher anatomical complexity around posterior points, as when manually annotated, the point where the lateral semicircular canal enters the vestibule is on an oblique plane. In addition, there is variation in the anatomic location of the confluence with the posterior semicircular canal, which can make this landmark susceptible to inter-observer variability.

The results of the additional validation in Figs 13 and 14 show that the majority of the prediction errors are distributed within a narrow band around the bias line indicating that the prediction errors are low. This is supported by the statistics in Table 4. The median distance between the predictions and ground truth is well below the suggested threshold of 0.5 mm [22], and this is statistically confirmed at the 5% significance level using the Wilcoxon Signed Rank test.

Although we showed that data augmentation is an effective method of supplementing the dataset, the number of patients used in the training and testing, as well as the additional validation is relatively low. Furthermore, the data used was fairly homogeneous (with no pathologies). Therefore, it is not clear how the models will perform when faced with larger variations of data. As such, more validation of the models on larger datasets that also include pathological cases needs to be conducted in the future. As observed in Copson et al. [8], one of the advantages of the landmarks under

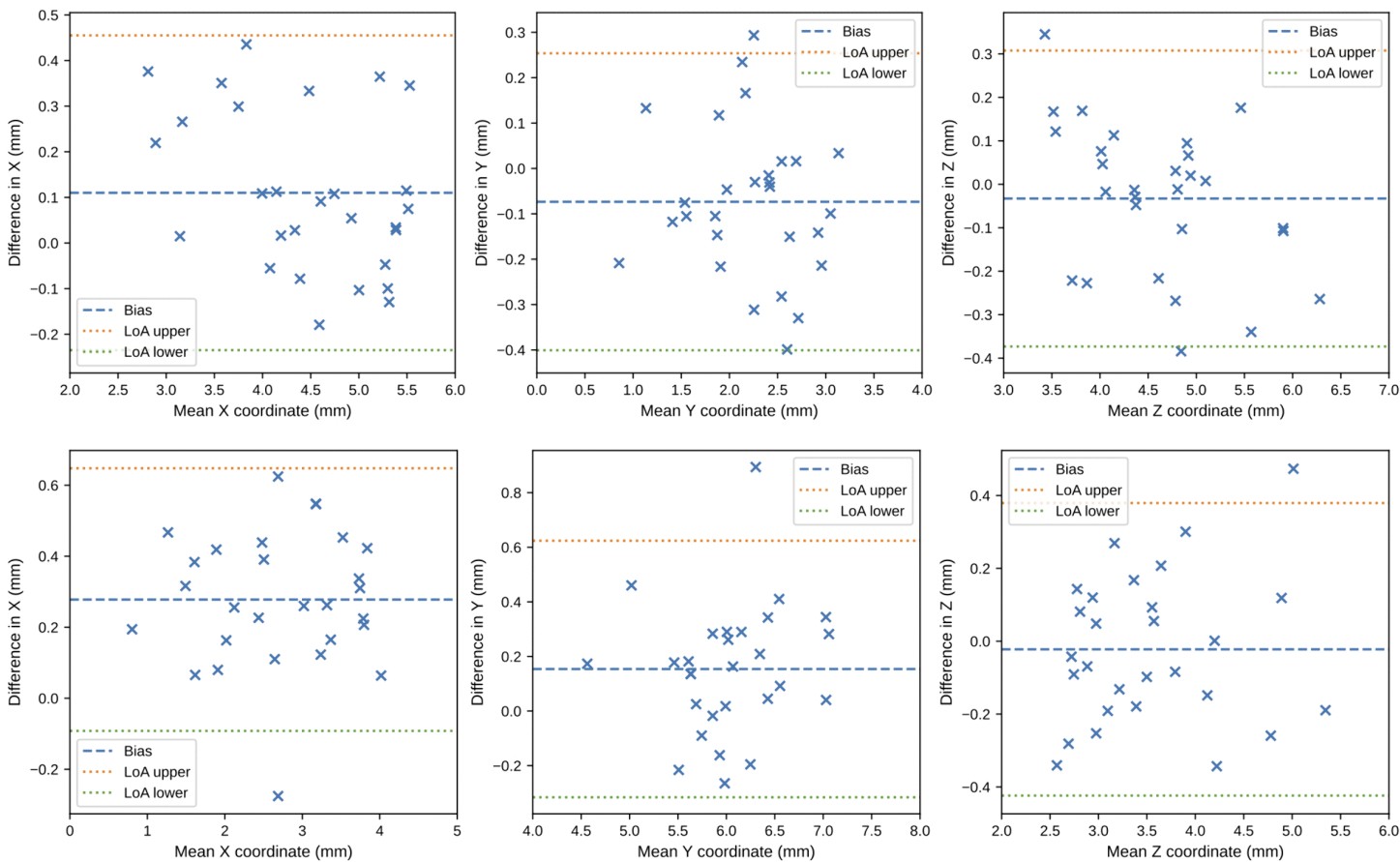

**Fig 14. Bland–Altman plots for U-Net with DSNT inference on the additional dataset.** Differences (in millimeters) between the model's predictions and the ground truth are shown across all three axes. The bias (mean difference) is shown as a blue dashed line, and the limits of agreement (mean ± 1.96 × SD) are shown as red and green lines.

**Table 4. Landmark localization results for the additional dataset. The performance indices shown are: median, minimum, maximum, first quartile (Q1), third quartile (Q3) and interquartile range (IQR). P-values < 0.05 provide evidence, at the 5% significance level, that the median error lies below 0.5 mm.**

| Metric | Down-Net (FC) | U-Net (DSNT) |
|---|---|---|
| Median | 0.322 | 0.378 |
| Minimum | 0.065 | 0.120 |
| Maximum | 1.264 | 0.903 |
| Q1-Q3 | 0.219-0.493 | 0.242-0.474 |
| IQR | 0.274 | 0.232 |
| P-Value | <0.001 | <0.001 |

consideration is that they are detectable in both CT and MRI. To capitalise on this, the proposed landmark detection framework should be adapted to detect landmarks in MRI as well. In this way, automatic landmark based registration methods can be developed for CT and MRI images of a patient. This will allow the combination of information available in both modalities for the purposes of diagnosis and surgical planning.

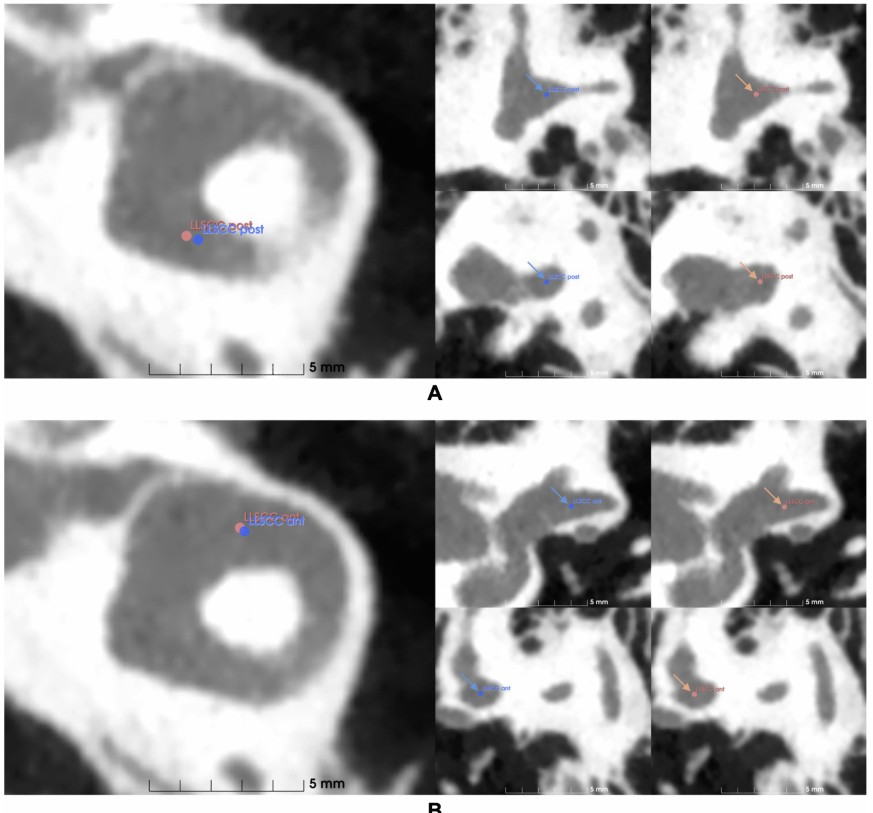

**Fig 15. Landmark localization results.** Blue and red points represent expert annotations and model predictions, respectively. The axial, coronal, and sagittal views are shown from left to right. **(A)** Anterior landmark with an error distance of 0.19 mm. **(B)** Posterior landmark with an error distance of 0.40 mm.

In addition, further refinement of the methods will be beneficial. For example, detection of the region of interest was performed using an exhaustive process and using one model (Down-Net). This may have influenced the results of the latter stages. Alternative methods such as the use of attention mechanisms, transformer-based localization, and adaptive region of interest detection may be useful in this application. An investigation into interpretability of the models will also be useful in future refinements.

## Conclusion

In this paper, we introduced a framework to detect landmarks of the lateral semi circular canals from Cone Beam CT scans of the temporal bone. We showed that this method can achieve a clinically acceptable level of accuracy compared to human annotations.

## Acknowledgments

This research was supported by The University of Melbourne's Research Computing Services and the Petascale Campus Initiative.

## Author contributions

**Data curation:** Bridget Copson.

**Formal analysis:** Zhixuan Wei.

**Methodology:** Zhixuan Wei, Sudanthi Wijewickrema.

**Project administration:** Zhixuan Wei, Sudanthi Wijewickrema.

**Resources:** Bridget Copson.

**Software:** Zhixuan Wei.

**Supervision:** Sudanthi Wijewickrema, Jean-Marc Gerard, Stephen O'Leary.

**Validation:** Zhixuan Wei, Sudanthi Wijewickrema.

**Visualization:** Zhixuan Wei.

**Writing – original draft:** Zhixuan Wei.

**Writing – review & editing:** Zhixuan Wei, Sudanthi Wijewickrema, Bridget Copson, Jean-Marc Gerard, Stephen O'Leary.

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
