## [Decision Letter · Decision Letter 0]

13 Oct 2025

PONE-D-25-27140A Deep Learning Framework for the Localization of Landmarks on the Lateral Semi Circular CanalsPLOS ONE

Dear Dr. Wei,

Thank you for submitting your manuscript to PLOS ONE. After careful consideration, we feel that it has merit but does not fully meet PLOS ONE’s publication criteria as it currently stands. Therefore, we invite you to submit a revised version of the manuscript that addresses the points raised during the review process.

Thank you for your patience in awaiting a decision on this manuscript, as it was difficult to obtain reviewers. Following my own review of the manuscript and reviewer comments, I am requesting Major Revisions on this manuscript. Please see comments below.  ==============================

We look forward to receiving your revised manuscript.

Kind regards,

Catalina I Villamil

Academic Editor

PLOS ONE

Journal Requirements:

3. We note you have included a table to which you do not refer in the text of your manuscript. Please ensure that you refer to Table 4 in your text; if accepted, production will need this reference to link the reader to the Table.

Additional Editor Comments:

Reviewer 1 provides very helpful advice regarding additional statistical information, tests, and parameters that should be reported. I further agree that, most importantly, the authors need to include an external validation dataset, as the original dataset contains only n =20 patients and was further subdivided for training and testing.

Reviewers' comments:

Reviewer's Responses to Questions

Comments to the Author

1. Is the manuscript technically sound, and do the data support the conclusions?

Reviewer #1: Partly

Reviewer #2: Yes

2. Has the statistical analysis been performed appropriately and rigorously? 

Reviewer #1: Yes

Reviewer #2: Yes

3. Have the authors made all data underlying the findings in their manuscript fully available?

Reviewer #1: No

Reviewer #2: Yes

4. Is the manuscript presented in an intelligible fashion and written in standard English?

Reviewer #1: Yes

Reviewer #2: Yes

5. Review Comments to the Author

Reviewer #1: This manuscript presents a deep learning framework for the localization of landmarks in the lateral semicircular canals using cone-beam CT data. The study addresses a clinically relevant and technically challenging task, and the authors demonstrate promising results showing that their system achieves accuracy comparable to expert performance and within surgical tolerance levels. The work is well structured, generally well written, and of potential value to the medical imaging and surgical planning communities.

Strengths:

1.The study focuses on an important clinical application with direct translational potential for skull base surgery.

2.The methodology is clear and logical, employing a stepwise multi-stage approach that balances computational efficiency with accuracy.

3.Comparative evaluation of several neural network architectures is a strong feature, providing a broader assessment of performance.

4.Use of multiple expert annotations for ground truth increases reliability.

5.The manuscript is clearly written and technically accessible.

Limitations and Points for Improvement:

1.Sample Size and Generalizability: The dataset consists of only 20 patients, all with normal anatomy. Even with augmentation, this raises concerns about robustness and generalizability. Inclusion of a larger and more heterogeneous dataset, ideally with pathological cases, would significantly strengthen the study.

2.External Validation: The absence of an external validation cohort is a major limitation. Testing on independent datasets is essential to demonstrate clinical applicability.

3.Statistical Rigor: While mean and standard deviation values are reported, the statistical analysis could be improved. Confidence intervals, hypothesis testing, or Bland–Altman plots would better substantiate the claim of equivalence to human experts.

4.ROI Determination: The exhaustive crop-size approach is functional but inefficient. Alternative approaches (e.g., attention mechanisms, transformer-based localization, or adaptive ROI detection) could be considered to modernize the framework.

5.Interpretability and Clinical Workflow: The manuscript would benefit from visualization of errors, attention maps, or examples of both successes and failures. Furthermore, details such as runtime per scan and workflow integration into surgical planning should be elaborated.

6.Data Availability: The dataset is not publicly available due to ethical restrictions, which limits reproducibility. While this is understandable, stronger justification or provision of anonymized subsets would help align with open science practices.

Reviewer #2: The precision the systems achieved to localize three-dimensional landmark coordinates of the semicircular canals is worthy of high praise.

According to the data shown in “[8]:Copson B, Wijewickrema S, Slinger C, Youssef D, Gerard JM, O’Leary S. Definition of a coordinate system for multi-modal images of the temporal bone and inner ear. Plos one. 2024”, intra and inter rater (experts) variation is very low. This means that it is an easy and consistent task for the “experts”. How do the authors see the value of this system.

6. PLOS authors have the option to publish the peer review history of their article (what does this mean?). If published, this will include your full peer review and any attached files.

Do you want your identity to be public for this peer review? For information about this choice, including consent withdrawal, please see our Privacy Policy.

Reviewer #1: Yes: Parisa Kaviani

Reviewer #2: No

---

## [Author Response · Author response to Decision Letter 1]

8 Dec 2025

Academic Editor’s comments

Please ensure that your manuscript meets PLOS ONE's style requirements, including those for file naming. The PLOS ONE style templates can be found here and here.

We have ensured that the manuscript uses the PLOS ONE template and meets the style requirements.

We note that you have indicated that there are restrictions to data sharing for this study. For studies involving human research participant data or other sensitive data, we encourage authors to share de-identified or anonymized data. However, when data cannot be publicly shared for ethical reasons, we allow authors to make their data sets available upon request. For information on unacceptable data access restrictions, please see here.

The data collected is owned by the Royal Victorian Eye and Ear Hospital, Melbourne, Australia. The ethics under which the data was collected by the University of Melbourne researchers restricts its public sharing. However, anonymised data can be shared on request. Please contact Prof Stephen O'Leary (sjoleary@unimelb.edu.au) the head of the Department of Otolaryngology in the University of Melbourne and also the owner of the ethics (#08/796H/18) of the data. Researchers can also contact Kerryn Baker (kerryn.baker@eyeandear.org.au), Education Precinct Manager of the Royal Victorian Eye and Ear Hospital, about the ethics.

We note you have included a table to which you do not refer in the text of your manuscript. Please ensure that you refer to Table 4 in your text; if accepted, production will need this reference to link the reader to the Table.

Thank you for noticing this oversight. We have corrected this.

We feel that the reviewers’ comments are very constructive and neither has requested us to cite references that are not relevant.

Additional Editor Comments:

Reviewer 1 provides very helpful advice regarding additional statistical information, tests, and parameters that should be reported. I further agree that, most importantly, the authors need to include an external validation dataset, as the original dataset contains only n =20 patients and was further subdivided for training and testing.

We admit that n=20 is small. We have however taken steps (such as augmentation and cross validation) to address this. Also, please note that Reference #21 of the paper in the field have used similar (or lesser) number of patients. However, we have done further validation on 14 new patients. Please note that due to practical considerations such as data collection, labelling by experts and time constraints for the first author who is a PhD student in his final year, we have limited the external validation to 14 patients. Please note that we have called this ‘additional validation’ as the dataset was drawn from the same population as the training and test sets. The changes to the manuscript are in lines 296-316.

We agree that the additional statistical tests suggested will add value and have conducted them (lines 305–315, figures 13, 14 tables 4).

Reviewer 1’s comments

1. Sample Size and Generalizability: The dataset consists of only 20 patients, all with normal anatomy. Even with augmentation, this raises concerns about robustness and generalizability. Inclusion of a larger and more heterogeneous dataset, ideally with pathological cases, would significantly strengthen the study.

Thank you for the comments. We agree with the reviewer. Unfortunately, we have been limited in the number of patients due to practical issues in collecting and labelling data. We have taken measures such as augmentation to overcome issues of over-fitting and cross validation.

However, as per your suggestion, we have further conducted an external validation that addresses the issue of generalizability. Please note that we have called this ‘additional validation’ as the dataset was drawn from the same population as the training and test sets. Please see Section Additional Validation of the manuscript.

This patient cohort for this study were cochlear implant patients. As such, it is rare for any abnormalities and pathologies in and around the lateral semi-circular canals (where the landmarks are located) to be present. We plan to extend the method to include pathological cases in the future. This has been added as a limitation in the discussion (lines 342-345).

2.External Validation: The absence of an external validation cohort is a major limitation. Testing on independent datasets is essential to demonstrate clinical applicability.

Thank you. We acknowledge this is an issue and have conducted an additional validation (Section Additional Validation, Table 4 etc).

3.Statistical Rigor: While mean and standard deviation values are reported, the statistical analysis could be improved. Confidence intervals, hypothesis testing, or Bland–Altman plots would better substantiate the claim of equivalence to human experts.

Thank you for this suggestion. We have conducted these statistical analyses. Lines 305-315 and table 4, figures 13, 14 in the manuscript reflect these changes. Please note that we have used non-parametric statistics for this analysis as the number of patients in the additional validation set was 14 with no augmentations. This leads to the violation of the normality assumption required to conduct parametrics statistics.

4.ROI Determination: The exhaustive crop-size approach is functional but inefficient. Alternative approaches (e.g., attention mechanisms, transformer-based localization, or adaptive ROI detection) could be considered to modernize the framework.

We agree that this approach of crop size detection is not the most optimal. We will conduct more efficient crop size detection methods such as those suggested by the reviewer in future work (lines 356-357).

5.Interpretability and Clinical Workflow: The manuscript would benefit from visualization of errors, attention maps, or examples of both successes and failures. Furthermore, details such as runtime per scan and workflow integration into surgical planning should be elaborated.

We have added figures to visualise results including errors (Figures 15).

We did not include run time per scan as it is a subjective measure depending on factors such as processing power and memory capacity of the computer.

Automating the identification of landmarks enables patient scans to be transformed into a standardized coordinate system, improving the clinical workflow by (1) facilitating automation of downstream tasks such as multi-modal registration, surgical planning, and pathology identification, and (2) minimizing the need for human intervention. This has been added to the manuscript (lines 318-322).

6.Data Availability: The dataset is not publicly available due to ethical restrictions, which limits reproducibility. While this is understandable, stronger justification or provision of anonymized subsets would help align with open science practices.

The data collected is owned by the Royal Victorian Eye and Ear Hospital, Melbourne, Australia. The ethics under which the data was collected by the University of Melbourne researchers restricts its public sharing. However, anonymised data can be shared on request. Please contact Prof Stephen O'Leary (sjoleary@unimelb.edu.au) the head of the Department of Otolaryngology in the University of Melbourne and also the owner of the ethics (#08/796H/18) of the data. Researchers can also contact Kerryn Baker (kerryn.baker@eyeandear.org.au), Education Precinct Manager of the Royal Victorian Eye and Ear Hospital, about the ethics.

Reviewer 2’s comments

According to the data shown in “[8]:Copson B, Wijewickrema S, Slinger C, Youssef D, Gerard JM, O’Leary S. Definition of a coordinate system for multi-modal images of the temporal bone and inner ear. Plos one. 2024”, intra and inter rater (experts) variation is very low. This means that it is an easy and consistent task for the “experts”. How do the authors see the value of this system?

Automating the identification of landmarks enables patient scans to be transformed into a standardized coordinate system, improving the clinical workflow by (1) facilitating automation of downstream tasks such as multi-modal registration, surgical planning, and pathology identification, and (2) minimizing the need for human intervention. This has been added to the manuscript (lines 318-322).

---

## [Decision Letter · Decision Letter 1]

17 Mar 2026

PONE-D-25-27140R1A Deep Learning Framework for the Localization of Landmarks on the Lateral Semi Circular CanalsPLOS One

Dear Dr. Wei,

Thank you for submitting your revised manuscript to PLOS ONE and thank you for your patience. I was able to obtain a review from one additional reviewer. I concur with the reviewer that the major issues with this manuscript (lack of external validation) was addressed. However, there are some minor comments to be addressed that will improve clarity of the figures and manuscript. Therefore, we invite you to submit a revised version of the manuscript that addresses the points raised during the review process.

We look forward to receiving your revised manuscript.

Kind regards,

Catalina I Villamil

Academic Editor

PLOS One

Journal Requirements:

Additional Editor Comments:

I would note that in Fig. 15 it is quite difficult to see the landmarks that are being shown. I would suggest making the dot that designates the landmark(s) bigger. I also suggest making this figure colorblind friendly (e.g. red and blue instead of red and green).

Reviewers' comments:

Reviewer's Responses to Questions

Comments to the Author

1. If the authors have adequately addressed your comments raised in a previous round of review and you feel that this manuscript is now acceptable for publication, you may indicate that here to bypass the “Comments to the Author” section, enter your conflict of interest statement in the “Confidential to Editor” section, and submit your "Accept" recommendation.

Reviewer #3: (No Response)

2. Is the manuscript technically sound, and do the data support the conclusions?

Reviewer #3: Yes

3. Has the statistical analysis been performed appropriately and rigorously? 

Reviewer #3: Yes

4. Have the authors made all data underlying the findings in their manuscript fully available?

Reviewer #3: Yes

5. Is the manuscript presented in an intelligible fashion and written in standard English?

Reviewer #3: Yes

6. Review Comments to the Author

Reviewer #3: Thank you for the opportunity to review the manuscript entitled “A Deep Learning Framework for the Localization of Landmarks on the Lateral Semicircular Canals.” This study presents an innovative application of deep learning to a longstanding challenge in otologic and vestibular research. Automated and reproducible localization of canal landmarks has potential to advance both clinical and morphometric investigations of the inner ear.

Visualizing the detailed morphology of the endosseous and membranous labyrinths remain one of the biggest challenges in ontological research. In this context, the authors’ effort to develop a computational framework for landmark detection is both appropriate and potentially impactful.

While the deep learning architecture and training procedures fall outside my primary area of expertise, I am happy to provide some feedback on the morphometric study of the LSCC. Below, I offer several suggestions focused on the morphometric and anatomical aspects of the study.

• In your “Landmarks of the LSCC” section, you need a clearer definition of where each landmark was placed in reference to anatomical features (e.g., “middle of the bony ampulla and point at which the slender LSCC joins the vestibule”). Please add in this description of the features.

• Line 215: “Fi” should be “Fig”.

• Line 298: What is meant by the word “annotated”?

• Lines 300 and 313: You say the upper boundary for the clinically significant distance threshold is 0.5mm and cite Mueller et al., 2021. That paper reported ~0.5mm as the maximum distance found between the facial recess trajectory orientation to the facial nerve and osseous spiral lamina. However, the average distances they report are 0.44mm, and 0.35mm, respectively. I would recommend elaborating here on why exactly 0.5mm is the best fit for establishing your upper boundary?

• Line 331: What is meant by “ground truth”?

• Line 332: “…coordinates selected by human experts (citation needed)”

• Figure 13. It seems the greater variability in the posterior landmark is driven by one individual. Can you elaborate on why this may be? It might be worth doubling checking the scan to see what is going on here.

7. PLOS authors have the option to publish the peer review history of their article (what does this mean?). If published, this will include your full peer review and any attached files.

Do you want your identity to be public for this peer review? For information about this choice, including consent withdrawal, please see our Privacy Policy.

Reviewer #3: Yes: Christopher M Smith

---

## [Author Response · Author response to Decision Letter 2]

22 Apr 2026

Dear Editor,

Thank you for giving us the opportunity to submit a revised version of our manuscript titled “A Deep Learning Framework for the Localization of Landmarks on the Lateral Semi Circular Canals” to PlosONE. We appreciate the time and effort that you and the reviewers have dedicated to providing feedback on our manuscript. We have modified the manuscript to reflect the suggestions provided. We have highlighted the changes within the manuscript. The manuscript has also been updated with the new LaTeX template available on the PlosONE website.

Here is a point-by-point response to the reviewers’ comments and concerns. All the responses are in italic and the corresponding changes in the manuscript are referred to in this rebuttal letter using line numbers.

Journal Requirements:

Not relevant

Thank you for the reviewer’s point, we have added a new reference to elaborate why we chose 0.5mm as the upper boundary for the statistical comparison. The new reference is: Schneider D, Hermann J, Mueller F, Braga GO'TB, Anschuetz L, Caversaccio M, Nolte L, Weber S, Klenzner T. Evolution and Stagnation of Image Guidance for Surgery in the Lateral Skull: A Systematic Review 1989–2020. Front Surg. 2021;7:604362. (line 308)

Additional Editor Comments:

I would note that in Fig. 15 it is quite difficult to see the landmarks that are being shown. I would suggest making the dot that designates the landmark(s) bigger. I also suggest making this figure colorblind friendly (e.g. red and blue instead of red and green).

Thank you for pointing this out. We have updated Fig 15 using bigger landmarks with the colors red and blue. (Fig 15)

Reviewer 3’s comments

Thank you for the opportunity to review the manuscript entitled “A Deep Learning Framework for the Localization of Landmarks on the Lateral Semicircular Canals.” This study presents an innovative application of deep learning to a longstanding challenge in otologic and vestibular research. Automated and reproducible localization of canal landmarks has potential to advance both clinical and morphometric investigations of the inner ear.

Visualizing the detailed morphology of the endosseous and membranous labyrinths remain one of the biggest challenges in ontological research. In this context, the authors’ effort to develop a computational framework for landmark detection is both appropriate and potentially impactful.

While the deep learning architecture and training procedures fall outside my primary area of expertise, I am happy to provide some feedback on the morphometric study of the LSCC. Below, I offer several suggestions focused on the morphometric and anatomical aspects of the study.

• In your “Landmarks of the LSCC” section, you need a clearer definition of where each landmark was placed in reference to anatomical features (e.g., “middle of the bony ampulla and point at which the slender LSCC joins the vestibule”). Please add in this description of the features.

Thank you for your comment. We have added a clearer definition in the section “Landmarks of the LSCC”. The Anterior Landmark was defined as the centre-point of the junction between the anterior aspect of the LSCC, and the utricle - that is, the point at which the slender bony canal opens into the vestibule anteriorly - confirmed in all three imaging planes. The Posterior Landmark was defined equivalently at the posterior junction of the LSCC and the utricle, where the canal re-enters the vestibule posteriorly, again confirmed in all three planes. (line 104 to 109)

• Line 215: “Fi” should be “Fig”.

Fixed, thank you. (line 220)

• Line 298: What is meant by the word “annotated”?

Thank you for asking, it is a term used in deep learning which represents the process used to label the ground truth. For example, in our automatic landmarks detection task, it means the ground truth landmarks of the 14 additional patients’ scans were manually identified by expert clinicians. (No change in the manuscript)

• Lines 300 and 313: You say the upper boundary for the clinically significant distance threshold is 0.5mm and cite Mueller et al., 2021. That paper reported ~0.5mm as the maximum distance found between the facial recess trajectory orientation to the facial nerve and osseous spiral lamina. However, the average distances they report are 0.44mm, and 0.35mm, respectively. I would recommend elaborating here on why exactly 0.5mm is the best fit for establishing your upper boundary?

Thank you for pointing this out. We agree that the literature Mueller et al., 2021 is insufficient for the choice of the upper boundary 0.5mm. We have cited another paper Schneider et. al., which clearly suggested it as the threshold for lateral skull based image guidance. The elaboration has been added in the manuscript: Accuracy requirements in image guided surgery and image registration using landmarks are not standardised, and reasonable thresholds depend on the precision required by the surgical technique. For lateral skull base image guidance, a value of 0.5mm has been suggested (Schneider et. al.), and we adopted this as the predefined upper boundary for evaluating the clinical acceptability of our landmark detection results. (line 305 to 309; line 321; line 350)

The reference added: Schneider D, Hermann J, Mueller F, Braga GO'TB, Anschuetz L, Caversaccio M, Nolte L, Weber S, Klenzner T. Evolution and Stagnation of Image Guidance for Surgery in the Lateral Skull: A Systematic Review 1989–2020. Front Surg. 2021;7:604362.

• Line 331: What is meant by “ground truth”?

Thank you for asking, “ground truth” is a term used in supervised deep learning which represents the target of the model’s optimization direction. In our task it is the median of the landmarks coordinates picked by the experts. The model’s training process is the optimization to minimize the distance between the predicated landmarks and the ground truth landmarks. (No change in the manuscript)

• Line 332: “…coordinates selected by human experts (citation needed)”

Thanks for the comment. We have added the citation. (line 336)

• Figure 13. It seems the greater variability in the posterior landmark is driven by one individual. Can you elaborate on why this may be? It might be worth doubling checking the scan to see what is going on here.

Thank you for pointing this out. We have checked the scans of the greatest difference shown in Figure 13’s posterior landmarks. The scans are not from a single patient, they come from three different patients. As far as we can see, there is no obvious pattern in these three cases. We have also observed the slightly higher variation in the posterior landmark coordinates. This is likely because of higher variations in the human landmark detections which were used as ground truth for training our models. We have already addressed this in the discussion. The reason behind the differences in human landmarks detections for anterior and posterior points may be the higher anatomical complexity around posterior points, as when manually annotated, the point where the lateral semicircular canal enters the vestibule is on an oblique plane. In addition, there is variation in the anatomic location of the confluence with the posterior semicircular canal, which can make this landmark susceptible to inter-observer variability. We have added this to the discussion for clarification (line 340 to 345).

---

## [Editor Report · Decision Letter 2]

24 Apr 2026

A Deep Learning Framework for the Localization of Landmarks on the Lateral Semi Circular Canals

PONE-D-25-27140R2

Dear Dr. Wei,

We’re pleased to inform you that your manuscript has been judged scientifically suitable for publication and will be formally accepted for publication once it meets all outstanding technical requirements.

Kind regards,

Catalina I Villamil

Academic Editor

PLOS One

---

## [Editor Report · Acceptance letter]

PONE-D-25-27140R2

PLOS One

Dear Dr. Wei,

I'm pleased to inform you that your manuscript has been deemed suitable for publication in PLOS One. Congratulations! Your manuscript is now being handed over to our production team.

Kind regards,

on behalf of

Dr. Catalina I Villamil

Academic Editor

PLOS One